# UDM-GRPO: Stable and Efficient Group Relative Policy Optimization for Uniform Discrete Diffusion Models

**Jiaqi Wang** [1 2 *]  **Haoge Deng** [2 *]  **Ting Pan** [2 *]  **Yang Liu** [2]
**Chengyuan Wang** [2]  **Fan Zhang** [2]  **Yonggang Qi** [1]  **Xinlong Wang** [2]

## Abstract

Uniform Discrete Diffusion Model (UDM) has recently emerged as a promising paradigm for discrete generative modeling; however, its integration with reinforcement learning remains largely unexplored. We observe that naively applying GRPO to UDM leads to training instability and marginal performance gains. To address this, we propose **UDM-GRPO**, the first framework to integrate UDM with RL. Our method is guided by two key insights: (i) treating the final clean sample as the action provides more accurate and stable optimization signals; and (ii) reconstructing trajectories via the diffusion forward process better aligns probability paths with the pretraining distribution. Additionally, we introduce two strategies, Reduced-Step and CFG-Free, to further improve training efficiency. **UDM-GRPO** significantly improves base model performance across multiple T2I tasks. Notably, GenEval accuracy improves from $69\%$ to $96\%$ and PickScore increases from $20.46$ to $23.81$, achieving state-of-the-art performance in both continuous and discrete settings. On the OCR benchmark, accuracy rises from $8\%$ to $57\%$, further validating the generalization ability of our method. Code is available at https://github.com/Yovecent/UDM-GRPO.

## 1. Introduction

Recent advances in visual generative models have achieved remarkable generation quality (Lipman et al., 2022; Ho et al., 2020; Chang et al., 2022). In parallel, Uniform Discrete Diffusion (Gat et al., 2024; Wang et al., 2025a; Deng et al.,

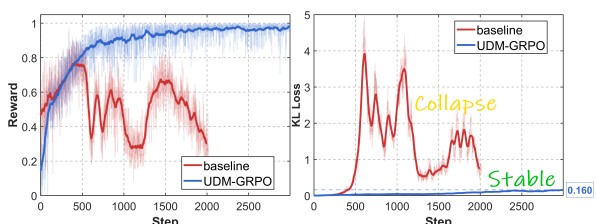

*Figure 1.* Reward–step training curve. The baseline suffers from optimization collapse after 500 steps, characterized by violent reward oscillation and exploding KL divergence. In contrast, our UDM-GRPO achieves stable convergence with sustained reward improvement and bounded KL loss.

2025) has emerged as a promising paradigm for discrete generation. By using parallel token updates and progressive refinement, it outperforms traditional mask-based methods (Xie et al., 2024). Despite these advances, pretrained models often struggle with tasks requiring precise alignment with human preferences (Lee et al., 2023) or complex compositional generation (Yan et al., 2025). Meanwhile, Reinforcement Learning (RL) (Sutton et al., 1998), particularly Group Relative Policy Optimization (GRPO) (Shao et al., 2024), has proven highly effective in enhancing the reasoning capabilities of Large Language Models (LLMs) (Guo et al., 2025). Motivated by this success, recent works have extended GRPO to visual generation (Xue et al., 2025; Wang et al., 2025b). Notably, approaches like Flow-GRPO (Liu et al., 2025) have demonstrated substantial gains by formulating the denoising process as a policy optimization problem. However, the integration of RL into Uniform Discrete Diffusion remains largely unexplored. This work takes the first step toward bridging this gap.

Drawing inspiration from Flow-GRPO, we first construct a baseline adaptation for Uniform Discrete Diffusion. To circumvent the non-differentiability inherent in the discretized sampling process, we define the policy action as the intermediate predicted sample at each timestep, theoretically aligning the optimization objective with the Flow-GRPO framework. However, this direct adaptation proves fundamentally unstable. As shown in Figure 1 (red curve), the model achieves transient gains during the first 500 steps. It then collapses catastrophically, exhibiting violent reward oscillations and an exploding KL divergence. We attribute this instability to two critical misalignments: **(i) Inaccurate intermediate actions.** Early-step predictions are high-entropy

*Equal contribution . This work was done at Beijing Academy of Artificial Intelligence. [1]Beijing University of Posts and Telecommunications [2]Beijing Academy of Artificial Intelligence. Correspondence to: Xinlong Wang <xinlong.wang96@gmail.com>, Yonggang Qi <qiyg@bupt.edu.cn>.

*Proceedings of the 43rd International Conference on Machine Learning*, Seoul, South Korea. PMLR 306, 2026. Copyright 2026 by the author(s).

and inaccurate, so treating them as actions forces the model to learn from noisy and incorrect signals; **(ii) Biased distribution of backward trajectory.** Optimizing on the reverse process induces a distribution shift from the forward process during pretraining. This discrepancy biases the learned probability path, effectively leading to out-of-distribution (OOD) training (Liu et al., 2021).

To address these challenges, we introduce UDM-GRPO, the first framework integrating Uniform Discrete Diffusion with GRPO for text-to-image generation. To ensure optimization stability, we employ two core strategies: (1) defining the policy action as the final clean sample—rather than intermediate noisy predictions—to guarantee reward-consistent optimization; and (2) reconstructing training trajectories via the forward process to eliminate sampling-induced distribution shifts.

We further introduce two strategies to enhance the efficiency of training. To mitigate the slow convergence caused by gradient dispersion across multi-step optimization, we propose a Reduced-Step training strategy that concentrates optimization on critical high-noise timesteps. Additionally, a CFG-Free scheme is adopted to avoid simultaneous optimization of conditional and unconditional objectives, substantially reducing computational overhead.

The improvement of our methods is evident in Figure 1. In contrast to the baseline, UDM-GRPO demonstrates a stable and sustained increase in reward without collapse, while maintaining a low and bounded KL divergence, validating the robustness of our framework.

Our contributions are summarized as follows: (1) We propose the first method to integrate GRPO into Uniform Discrete Diffusion for T2I tasks. UDM-GRPO addresses the instability of direct integration by unifying the action across timesteps as the final clean sample and reconstructing the training trajectory via the forward process. (2) We propose two strategies to improve training efficiency: Reduced-Step optimization and CFG-Free training. (3) Extensive experiments across multiple benchmarks validate the effectiveness of our approach. In particular, UDM-GRPO enables the base model URSA to achieve state-of-the-art performance on GenEval (Ghosh et al., 2024) and PickScore (Kirstain et al., 2023) for both discrete and continuous generation.

## 2. Related work

**Discrete Diffusion Model** Diffusion models have achieved remarkable success in continuous domains, demonstrating strong sample quality and scalability for visual synthesis (Labs, 2024; Seedream et al., 2025; Brooks et al., 2024; Wan et al., 2025; Cheng et al., 2026). Extending diffusion to discrete domains introduces challenges due to the categorical nature of discrete variables. Early works (Austin

et al., 2021; Hoogeboom et al., 2021) formalized diffusion over categorical spaces via multinomial transitions and discrete denoising objectives. Building on these foundations, one line of work adopts masked image modeling (MIM) for discrete image generation through iterative masked token prediction (Chang et al., 2022; 2023; Xie et al., 2024; Bai et al., 2024; Hong et al., 2022; Yu et al., 2023), showing strong performance with efficient parallel decoding. More recently, uniform discrete diffusion (Gat et al., 2024) has emerged as a simplified formulation by explicitly parameterizing a time-dependent categorical corruption process. Fudoki (Wang et al., 2025a) and Next-Omni (Luo et al., 2025a) integrate this framework into unified models for image generation, while URSA (Deng et al., 2025) demonstrates competitive or superior performance to continuous diffusion on both image and video benchmarks. In this work, we adopt URSA as our baseline, as it provides a strong and representative implementation of uniform discrete diffusion for image generation.

**Reinforcement Learning in Text-to-Image Generation** Reinforcement learning has become a key research direction for aligning text-to-image models with human preferences through feedback signals. Existing approaches can be broadly divided into two paradigms: (1) Direct Preference Optimization (Rafailov et al., 2023) casts alignment as a preference classification task over ranked output pairs, allowing direct policy updates without explicit reward modeling (Wallace et al., 2024; Deng et al., 2024a; Yang et al., 2024). (2) Policy-based RL methods. Early efforts in this line primarily adopt Proximal Policy Optimization (PPO) (Schulman et al., 2017). DDPO (Black et al., 2023) formulates diffusion denoising as a multi-step Markov Decision Process, enabling RL beyond likelihood maximization. Following its success in large language models, GRPO (Shao et al., 2024) has been extended to visual generation, including autoregressive models (Wang et al., 2025b), mask-based diffusion (Luo et al., 2025b), continuous diffusion models and flow-matching (Xue et al., 2025; Liu et al., 2025; Li et al., 2025; He et al., 2025). However, stable and effective RL for Uniform Diffusion remains underexplored. Building on this progress, we introduce GRPO to Uniform Diffusion. We observe that a direct adaptation of the Flow-GRPO formulation leads to severe training instability. To address these challenges, we propose UDM-GRPO, the first framework that enables stable and efficient reinforcement learning for Uniform Discrete Diffusion.

## 3. Initial Exploration

Uniform Discrete Diffusion has emerged as a robust paradigm for discrete generation. However, standard training relies on supervised cross-entropy minimization, which limits its ability to optimize complex, non-differentiable

objectives or handle intricate generation tasks. Therefore, we adopt GRPO (Shao et al., 2024) to solve these limitations. Motivated by the success of Flow-GRPO (Liu et al., 2025), we explore a direct integration of the Flow-GRPO framework with UDM, as described in this section. Specifically, we first review the fundamentals of Uniform Discrete Diffusion and the Flow-GRPO formulation in Sections 3.1 and 3.2, respectively, and then introduce our preliminary approach to combine the two in Section 3.3.

### 3.1. Uniform Discrete Diffusion

Discrete Flow Matching (DFM)/Diffusion (Gat et al., 2024; Shaul et al., 2024) is a class of generative models that transport a source distribution $p_0(x)$ to a target data distribution $p_1(x)$ on discrete state spaces $\mathcal{S} = [K]^D$, where $[K] = \{1, \ldots, K\}$ denotes the vocabulary and $D$ is the sequence length. In contrast to masking-based diffusion, which typically performs non-refinable local generation, uniform discrete diffusion starts from categorical noise and iteratively refines all tokens, enabling higher-fidelity synthesis. Specifically, *Uniform Discrete Diffusion Model* (UDM) specifies $p_0(x) \sim Unif([K])^D$ as the uniform distribution over the vocabulary and generates samples from $p_1(x)$ by jointly updating all tokens across timesteps, which has gradually attracted attention due to its particularly high generation quality.

**Probability paths.** To connect $p_0(x)$ and $p_1(x)$, DFM defines continuous intermediate distributions $\{p_t(x)\}_{t \in [0,1]}$,

$$p_t(x) \triangleq \sum_{x_1 \in \mathcal{S}} p_t(x \mid x_1)\, p_1(x_1), \qquad (1)$$

where $p_t(x \mid x_1)$ is the conditional forward distribution.

**Probability velocities.** To traverse the probability path $\{p_t(x)\}$, we model the generation process as a continuous-time Markov chain (CTMC) driven by a time-dependent probability velocity $u_t$, which guides the state from $x_t$ toward the terminal state $x_1$. For a small step size $\Delta t$, the state update rule is

$$x_{t+\Delta t} \sim \delta_{x_t}(\cdot) + \Delta t\, u_t(\cdot \mid x_t). \qquad (2)$$

**Training.** The model is trained to predict the original data $x_1 \sim p_1(x)$ from the noised data $x_t \sim p_t(x \mid x_1)$ by minimizing the cross-entropy objective:

$$\mathcal{L}_{\text{CE}}(\theta) = \mathbb{E}_{t \sim \mathcal{U}[0,1],\, x_1,\, x_t} \big[ -\log p_\theta(x_1 \mid x_t) \big]. \qquad (3)$$

**Inference.** Although sampling can theoretically be performed according to Eq. 2, in practice UDM adopts an Euler solver with a two-stage conditional sampling scheme for efficient generation (Shaul et al., 2024). Specifically, given $x_t$,

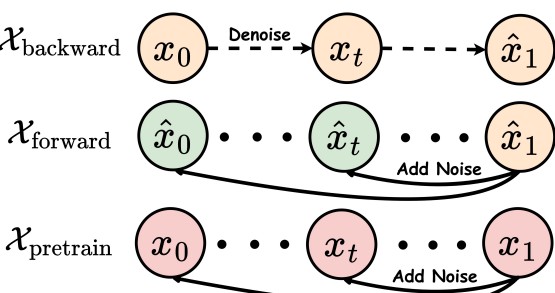

*Figure 2.* Illustration of the three trajectories. $\mathcal{X}_{\text{backward}}$ denoises $x_0$ via the reverse process to obtain $\hat{x}_1$. In contrast, $\mathcal{X}_{\text{forward}}$ and $\mathcal{X}_{\text{pretrain}}$ share the same forward diffusion process but differ in their clean sources: $\hat{x}_1$ for $\mathcal{X}_{\text{forward}}$ and $x_1$ from the pretraining dataset for $\mathcal{X}_{\text{pretrain}}$, resulting in $\hat{x}_t$ and $x_t$, respectively.

we first sample an intermediate prediction $x_1^t \sim p_\theta(\cdot \mid x_t)$ from the model trained under Eq. 3, and then update the state through a parameter-free rule-based mapping:

$$x_{t+\Delta t} \sim \delta_{x_t}(\cdot) + \Delta t\, u_t(\cdot, x_t \mid x_1^t), \qquad (4)$$

where $u_t(\cdot, x_t \mid x_1^t)$ denotes the conditional probability velocity.

**Trajectory Definition.** Given a caption–image pair $(c, x_1)$ from the pretraining dataset, we define three trajectories over $t \in [0, 1]$, using the reverse-process timesteps: (i) the backward trajectory $\mathcal{X}_{\text{backward}} = \{x_t\}_{t=0}^1$, where $x_t \sim p_\theta(x_{t+\Delta t} \mid x_t, c)$ is generated by following the reverse process, and $x_1$ denotes the model's estimate $\hat{x}_1$; (ii) the pretraining trajectory $\mathcal{X}_{\text{pretrain}} = \{x_t\}_{t=0}^1$, where $x_t \sim p_t(x \mid x_1)$ is generated by the forward diffusion process; (iii) the forward-process-based trajectory $\mathcal{X}_{\text{forward}} = \{\hat{x}_t\}_{t=0}^1$, where $\hat{x}_t \sim p_t(x \mid \hat{x}_1)$ is obtained by perturbing $\hat{x}_1$ via the same forward diffusion process. Figure 2 provides a detailed illustration of the three trajectories.

### 3.2. DDPO and Flow-GRPO

Most diffusion–RL methods are based on *Denoising Diffusion Policy Optimization* (DDPO) (Black et al., 2023), which formulates the reverse denoising process $\mathcal{X}_{\text{backward}}$ as a multi-step MDP. Formally, the induced MDP is $(\mathcal{S}, \mathcal{A}, \rho_0, P, R)$. At timestep $t$, the state is defined as $s_t \triangleq (c, t, x_t)$ where $c$ denotes the prompt and $x_t$ is the latent variable. The action corresponds to the denoised sample predicted by the model, $a_t \triangleq x_{t-1}$, and the policy is given by $\pi(a_t \mid s_t) \triangleq p_\theta(x_{t-1} \mid x_t, c)$. The transition is deterministic and specified by $P(s_{t+1} \mid s_t, a_t) \triangleq (\delta_c, \delta_{t-1}, \delta_{x_{t-1}})$, where $\delta_y$ denotes the Dirac delta distribution centered at $y$. The initial state distribution is $\rho_0(s_0) \triangleq (p(c), \delta_T, \mathcal{N}(0, I))$ and the reward is terminal-only: $R(s_t, a_t) \triangleq r(x_0, c)$.

Following this formulation, Flow-GRPO (Liu et al., 2025) converts the ODE-based denoising dynamics into an SDE to integrate Flow Matching with GRPO (Shao et al., 2024).

Specifically, given a prompt $c$, the model generates $G$ trajectories $\{\tau^i\}_{i=1}^G$, where $\tau^i = (x_0^i, x_{\Delta t}^i, \ldots, x_1^i)$ and $|\tau^i| = T$. The group-normalized advantage for the $i$-th trajectory is then computed as:

$$\hat{A}_i = \frac{R(x_1^i, c) - \mathrm{mean}\big(\{R(x_1^i, c)\}_{i=1}^G\big)}{\mathrm{std}\big(\{R(x_1^i, c)\}_{i=1}^G\big)} \tag{5}$$

Accordingly, Flow-GRPO optimizes the policy model by maximizing the following objective:

$$J(\theta) = \mathbb{E}_{c \sim \mathcal{C}, \{\tau^i\}_{i=1}^G \sim \pi_{\theta_{\mathrm{old}}}(\cdot | c)}$$
$$\left[ \frac{1}{G} \sum_{i=1}^G \frac{1}{T} \sum_{t=0}^1 \mathcal{J}_{policy}^{(t,i)} - \beta \, D_{\mathrm{KL}}\big(\pi_\theta \,\|\, \pi_{\mathrm{ref}}\big) \right] \tag{6}$$

where

$$\mathcal{J}_{policy}^{(t,i)} = \min\Big( r_t^i(\theta)\hat{A}_i, \, \mathrm{clip}(r_t^i(\theta), 1 - \epsilon, 1 + \epsilon)\hat{A}_i \Big)$$

$$r_t^i(\theta) = \frac{p_\theta(x_{t+\Delta t}^i \mid x_t^i, c)}{p_{\mathrm{old}}(x_{t+\Delta t}^i \mid x_t^i, c)}$$

### 3.3. Pilot Integration of GRPO and Uniform Discrete Diffusion

In our early study, we explored adapting Flow-GRPO to Uniform Discrete Diffusion, which focuses on the reverse sampling process $\mathcal{X}_{\mathrm{backward}}$. Since the optimization objective remains unchanged, the main challenge is how to calculate the transition probability $p_\theta(x_{t+\Delta t} \mid x_t, c)$ under Uniform Discrete Diffusion.

Recalling the Euler solver described in Section 3.1, we derive the following two properties. First, the transition from $x_t$ to $x_{t+\Delta t}$ requires sampling the intermediate prediction $x_1^t$; however, this non-differentiable step blocks gradient propagation from $x_t$ to $x_{t+\Delta t}$. Second, conditioned on $x_1^t$, the distribution of $x_{t+\Delta t}$ is uniquely determined by a fixed, parameter-free mapping. Consequently, the generation of $x_{t+\Delta t}$ is entirely governed by $x_1^t$, implying that learning $p_\theta(x_{t+\Delta t} \mid x_t, c)$ is effectively equivalent to learning $p_\theta(x_1^t \mid x_t, c)$.

Based on the above considerations, we redefined the action and policy as

$$a_t \triangleq x_1^t, \qquad \pi_\theta(a_t \mid s_t) \triangleq p_\theta(x_1^t \mid x_t, c). \tag{7}$$

This formulation preserves differentiability, retains Euler sampling efficiency, and enables policy optimization.

In discrete diffusion models, the network outputs logits directly, which enables more convenient and accurate probability computation. Specifically, the latent state $x_t \in \{1, \ldots, K\}^{B \times D}$ is a sequence of discrete tokens, where

$D$ denotes the number of tokens and satisfies $D = H \times W$ in the continuous formulation. The model outputs per-token logits $p_\theta(\cdot \mid x_t) \in \mathbb{R}^{B \times D \times K}$. Given the intermediate predicted samples $x_1^t \in \{1, \ldots, K\}^{B \times D}$, the policy probability can be computed as

$$\pi_\theta(a_t \mid s_t) = \prod_{\ell=1}^D \mathrm{Softmax}\Big( p_\theta(\cdot \mid x_t)_{:,\ell,:} \Big) \big[ x_{1,\ell}^t \big]. \tag{8}$$

Here, $p_\theta(\cdot \mid x_t)_{:,\ell,:}$ denotes the logits at position $\ell$, and $\big[ x_{1,\ell}^t \big]$ indicates indexing by the sampled token at $\ell$.

By redefining the action as $x_1^t$ and choosing to optimize along the $\mathcal{X}_{\mathrm{backward}}$ trajectory, we implement a preliminary integration of GRPO with UDM, which empirically improves the performance of the base model (Table 3).

## 4. Method

In this section, we first analyze the limitations that arise from a naive integration of Uniform Diffusion with Flow-GRPO in Section 4.1. We then propose **UDM-GRPO** in Section 4.2 to address these limitations. Finally, we present training acceleration strategies in Sections 4.3 and 4.4, including Reduced-Step training and CFG-Free training, respectively.

### 4.1. Instability Challenges of Uniform Diffusion under GRPO

As described in Section 3, we integrate Uniform Diffusion with GRPO by explicitly redefining the transition probability. However, as illustrated by the red curve in Figure 1, the reward initially increases during the first 500 training steps but soon exhibits severe fluctuations, while the KL divergence grows sharply, leading to unstable training dynamics and degraded performance. Further analysis reveals that this instability arises from the following two factors:

**Problem I: Inaccurate Intermediate Actions.** As shown in Figure 4, we visualize the entropy of the model output $p_\theta(\cdot \mid x_t)$ for $\mathcal{X}_{\mathrm{backward}}$ and the corresponding predictions $x_1^t$ at different timesteps (first row). Early stages exhibit high entropy, reflecting inherent uncertainty and yielding incoherent and noisy predictions. As the diffusion process proceeds, the entropy gradually decreases, and the model eventually predicts a clean sample $\hat{x}_1$. However, in our current RL-based formulation, each intermediate prediction $x_1^t$ is treated as the action. Although a positive advantage $A > 0$ is induced by the accurate final prediction $\hat{x}_1$, maximizing $p_\theta(x_1^t \mid x_t)$ compels the model to imitate unreliable intermediate predictions $x_1^t$ at early timesteps. As a result, the model learns misleading information, which destabilizes the training process and can ultimately lead to collapse.

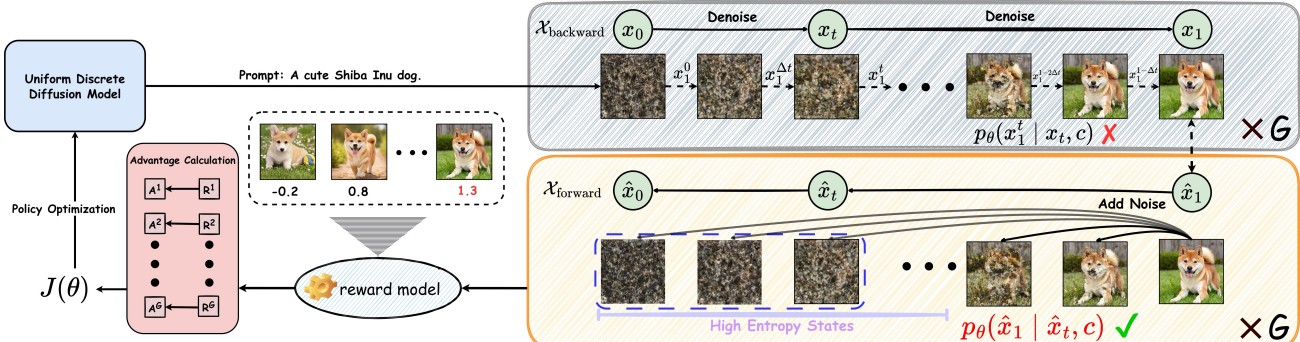

*Figure 3.* Overview of UDM-GRPO. Given a prompt, we first sample $G$ clean images $\hat{x}_1$ using the reverse process of UDM. To solve the instability caused by directly using this $\mathcal{X}_{\text{backward}}$ as trajectory and $x_1^t$ as action, we construct the training trajectory $\mathcal{X}_{\text{forward}}$ by perturbing $\hat{x}_1$ with forward process at different timesteps. Then we use $\mathcal{X}_{\text{forward}}$ as trajectory and $\hat{x}_1$ as action to calculate the transition probability $p_\theta(\hat{x}_1 \mid \hat{x}_t)$. Finally, we compute the advantage from rewards of $\hat{x}_1$ and optimize the policy model using the GRPO loss. Moreover, to improve training efficiency, we adopt the CFG-Free Strategy during sampling and the Reduced-Step Strategy to select early timesteps for policy optimization (blue dashed box).

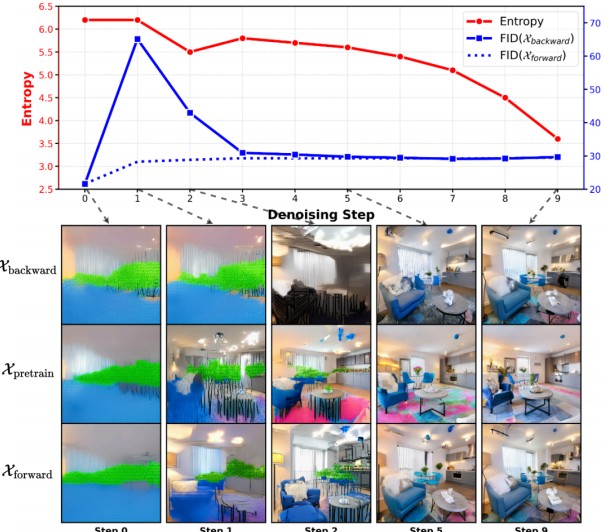

*Figure 4.* (i) The entropy of $p_\theta(\cdot \mid x_t)$ along the $\mathcal{X}_{\text{backward}}$ trajectory, and the FID between $\mathcal{X}_{\text{backward}}$ and $\mathcal{X}_{\text{pretrain}}$ as well as between $\mathcal{X}_{\text{forward}}$ and $\mathcal{X}_{\text{pretrain}}$ at different denoising timesteps (top). (ii) Visual comparison of the predicted $x_1^t$ images: $\mathcal{X}_{\text{backward}}$ (first row), $\mathcal{X}_{\text{pretrain}}$ (second row), and $\mathcal{X}_{\text{forward}}$ (third row).

**Problem II: Biased Distribution of Backward Trajectory.**
During pretraining, the model is trained to predict $x_1$ from the true forward process $\mathcal{X}_{\text{pretrain}}$, whereas during RL fine-tuning, optimization is performed on the model's own generated trajectory $\mathcal{X}_{\text{backward}}$. To investigate the discrepancy between these two distributions, we visualize the intermediate predictions $x_1^t$ in Figure 4. We observe a pronounced contrast: predictions from $\mathcal{X}_{\text{pretrain}}$ (second row) gradually become clearer starting from step 1, whereas those from $\mathcal{X}_{\text{backward}}$ (first row) remain significantly noisy during the early denoising steps. This performance gap indicates that $x_t \in \mathcal{X}_{\text{backward}}$ has drifted away from the training manifold $\mathcal{X}_{\text{pretrain}}$. Consequently, RL training exposes the model to out-of-distribution (OOD) (Liu et al., 2021) states, forcing it to learn from a biased probability trajectory.

## 4.2. UDM-GRPO

In this section, we propose UDM-GRPO, a new framework that resolves the aforementioned limitations.

**Key Insight I:** *To achieve more stable and precise optimization, it is desirable to select the accurate and reward-aligned denoised sample as the action.*

Standard diffusion pretraining inherently treats the clean image as the target for all $t$. Consistent with this principle, and given that our reward is defined solely on the final clean sample $\hat{x}_1$, we redefine the action at all timesteps in Eq. 7 to be $\hat{x}_1$:

$$a_t \triangleq \hat{x}_1, \qquad \pi(a_t \mid s_t) \triangleq p_\theta(\hat{x}_1 \mid x_t, c), \qquad (9)$$

This modification not only provides a more reward-consistent and precise optimization direction for RL, but also leads to further performance improvements (Table 3).

**Key Insight II:** *To mitigate distribution shift and preserve consistency with pretraining, the training trajectory should closely adhere to the forward diffusion process.*

Based on this, we adopt $\mathcal{X}_{\text{forward}}$ instead of $\mathcal{X}_{\text{backward}}$ as the training trajectory. We validate this choice by quantifying the discrepancy between the intermediate predictions of each trajectory and the pretraining distribution $\mathcal{X}_{\text{pretrain}}$ using Fréchet Inception Distance (FID) (Heusel et al., 2017). Detailed experimental settings are provided in Appendix B.

As illustrated in Figure 4, we can clearly see that $\mathcal{X}_{\text{forward}}$ yields consistently lower FID across all timesteps. In contrast, $\mathcal{X}_{\text{backward}}$ suffers from significant deviation caused by error accumulation in the early predictions, particularly in steps 1 and 2. The qualitative visualization further exhibits the same trend, with $\mathcal{X}_{\text{forward}}$ (third row) producing predictions that are visibly more consistent with $\mathcal{X}_{\text{pretrain}}$ than those

from $\mathcal{X}_{\text{backward}}$. These results indicate that $\mathcal{X}_{\text{forward}}$ is better aligned with the pretraining distribution $\mathcal{X}_{\text{pretrain}}$.

Through the above two modifications, we reformulate the original $T$-step MDP as follows:

$$
\begin{aligned}
s_t &\triangleq (\hat{x}_t, t, c), \quad a_t \triangleq \hat{x}_1, \quad \pi(a_t \mid s_t) \triangleq p_\theta(\hat{x}_1 \mid \hat{x}_t, c), \\
R(s, a) &\triangleq r(\hat{x}_1, c), \quad \rho_0(s) \triangleq Unif([K])^D.
\end{aligned}
\tag{10}
$$

Thus, the optimization objective remains the same as in Eq. 6, with the policy ratio now reformulated as:

$$
r_t^i(\theta) = \frac{p_\theta(\hat{x}_1^i \mid \hat{x}_t^i, c)}{p_{\theta_{\text{old}}}(\hat{x}_1^i \mid \hat{x}_t^i, c)}
\tag{11}
$$

As shown in Figure 1, our approach demonstrates significant advantages in convergence speed, performance, and stability, and Table 3 provides a more quantitative evaluation of these improvements. Compared to prior methods, it optimizes the forward process, naturally avoiding sampler constraints, high memory overhead, and inconsistencies with pretraining. Moreover, under this formulation, the policy loss for both diffusion models and LLMs can be viewed as an advantage-weighted version of their respective pre-trained losses, which further supports the validity and naturalness of our approach. A detailed overview of the framework is provided in Figure 3.

### 4.3. Reduced-Step Accelerated Optimization

Uniform optimization over all diffusion timesteps disperses gradients across the denoising trajectory, leading to inefficient convergence (He et al., 2025). As illustrated in Figure 4, the diffusion denoising process exhibits temporally decreasing stochasticity: early timesteps maintain high entropy, enabling extensive exploration of the state space, while later timesteps become more deterministic. The visualization of prediction $x_1^t$ for $\mathcal{X}_{\text{backward}}$ also shows that early denoising steps incur higher prediction errors, highlighting the need to focus optimization efforts on these stages. As a result, concentrating the optimization on steps with high noise leads to more significant gains. Motivated by this observation, we adopt a Reduced-Step training strategy to improve efficiency: for each sample $x_1$, we randomly select three consecutive timesteps from the first half of the diffusion timestep for training. As shown in Section 5.3, this approach significantly accelerates convergence.

### 4.4. CFG-Free Training

Most prior Diffusion–RL methods rely on classifier-free guidance (CFG) training, which jointly optimizes conditional and unconditional models and substantially increases training complexity. To simplify optimization, we eliminate CFG during training. Although this CFG-Free approach

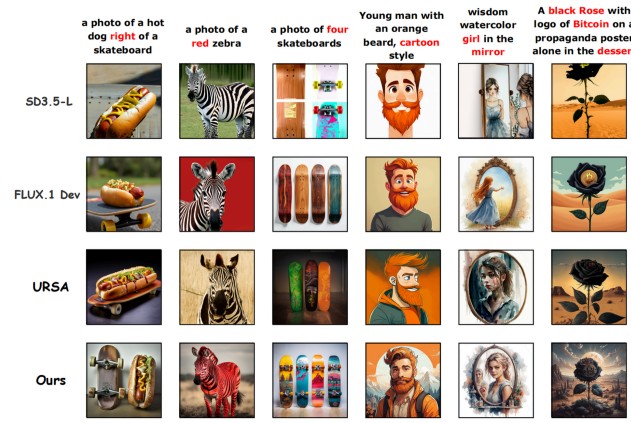

*Figure 5.* Qualitative Comparison. We evaluate our model against SD3.5-L, Flux.1 Dev and URSA using prompts from GenEval and PickScore, respectively.

initially degrades generation quality, the effect is transient: as training progresses, the model recovers and ultimately surpasses conventional CFG-based methods (Table 3).

## 5. Experiments

### 5.1. Experimental Settings

**Datasets and Reward Models.** Our experiments cover three tasks: *Compositional Image Generation*, *Visual Text Rendering*, and *Human Preference Alignment*. Compositional Image Generation evaluates the model's ability to understand and generate images with complex compositional constraints such as object count, color, and spatial relations. Visual Text Rendering focuses on accurately and consistently rendering text in realistic scenarios such as posters, advertisements, and books. Human Preference Alignment measures the alignment between generated images and human subjective preferences. For all tasks, we use the same datasets and corresponding reward models as in Flow-GRPO (Liu et al., 2025) for training and evaluation.

**Training and Evaluation.** We fine-tune a 1.7B-parameter text-to-image generation model pre-trained by URSA (Deng et al., 2025). Group sampling is adopted during training, with each batch comprising 16 groups of 8 image samples. We use AdamW optimizer (Loshchilov & Hutter, 2019) with $\beta_1 = 0.9$, $\beta_2 = 0.95$, a weight decay of 0.01, and a constant learning rate of 1e-6. We default to 10 inference steps for group sampling and 25 inference steps for evaluation. All experiments are conducted on 32 A100 (40GB) GPUs.

### 5.2. Main Results

To evaluate the effectiveness of our method, we integrate pre-trained URSA (Deng et al., 2025) text-to-image models with UDM-GRPO. As shown in Table 1, UDM-GRPO boosts the overall GenEval (Ghosh et al., 2024) score from 0.69 to 0.96,

*Table 1.* Comparison result on GenEval. Methods combined with GRPO are color-coded in gray. The best and second-best scores are marked in **bold** and underlined, respectively. Results for models other than ours are from (Liu et al., 2025) or their original papers.

| Model | #Params | Overall | Single Obj. | Two Obj. | Counting | Colors | Position | Attr. Binding |
|---|---|---|---|---|---|---|---|---|
| ▼ *Continuous models* | | | | | | | | |
| SD2.1 (Rombach et al., 2022) | 0.9B | 0.50 | 0.98 | 0.51 | 0.44 | 0.85 | 0.07 | 0.17 |
| SDXL (Podell et al., 2023) | 2.6B | 0.55 | 0.98 | 0.74 | 0.39 | 0.85 | 0.15 | 0.23 |
| SANA-1.5 4.8B (Xie et al., 2025) | 4.8B | 0.81 | 0.99 | 0.93 | 0.86 | 0.84 | 0.59 | 0.65 |
| NOVA (Deng et al., 2024b) | 1.4B | 0.71 | 0.99 | 0.91 | 0.62 | 0.85 | 0.33 | 0.56 |
| FLUX.1 Dev (Labs, 2024) | 12B | 0.66 | 0.98 | 0.81 | 0.74 | 0.79 | 0.22 | 0.45 |
| SD3.5-L (Esser et al., 2024) | 8B | 0.71 | 0.98 | 0.89 | 0.73 | 0.83 | 0.34 | 0.47 |
| SD3.5-M (Esser et al., 2024) | 2.5B | 0.63 | 0.98 | 0.78 | 0.50 | 0.81 | 0.24 | 0.52 |
| SD3.5-M (w/ Flow-GRPO) (Liu et al., 2025) | 2.5B | 0.95 | **1.00** | 0.99 | **0.95** | 0.92 | **0.99** | **0.86** |
| ▼ *Discrete models* | | | | | | | | |
| Emu3-Gen (Wang et al., 2024) | 8.5B | 0.54 | 0.98 | 0.71 | 0.34 | 0.81 | 0.17 | 0.21 |
| SimpleAR (Wang et al., 2025b) | 1.5B | 0.63 | - | 0.90 | - | - | 0.28 | 0.45 |
| MaskGen-XL (Kim et al., 2025) | 1.1B | 0.57 | 0.61 | 0.55 | 0.81 | 0.13 | 0.31 | 0.57 |
| Show-o (Xie et al., 2024) | 1.3B | 0.53 | 0.95 | 0.52 | 0.49 | 0.82 | 0.11 | 0.28 |
| Show-o (w/ Mask-GRPO) (Luo et al., 2025b) | 1.3B | 0.73 | 0.99 | 0.90 | 0.69 | 0.85 | 0.35 | 0.59 |
| FUDOKI (Wang et al., 2025a) | 1.5B | 0.77 | 0.96 | 0.85 | 0.56 | 0.88 | 0.68 | 0.67 |
| Emu3.5 (DiDA) (Cui et al., 2025) | 34B | 0.86 | - | - | - | - | - | - |
| URSA (Deng et al., 2025) | 1.7B | 0.69 | 0.99 | 0.91 | 0.60 | 0.87 | 0.28 | 0.49 |
| URSA (w/ UDM-GRPO) | 1.7B | **0.96** | **1.00** | **1.00** | **0.95** | **0.97** | 0.97 | 0.85 |

*Table 2.* Comparison results on GenEval, PickScore, and OCR. Our method, **UDM-GRPO**, is highlighted in gray. The best and second-best results are indicated by **bold** and underlined, respectively.

| Model | GenEval | PickScore | OCR |
|---|---|---|---|
| SDXL (Podell et al., 2023) | 0.55 | 22.42 | 0.14 |
| SD3.5-L (Esser et al., 2024) | 0.71 | 22.91 | **0.68** |
| FLUX.1-Dev (Labs, 2024) | 0.66 | 22.84 | 0.59 |
| URSA (Deng et al., 2025) | 0.69 | 21.79 | 0.08 |
| URSA (w/o CFG) (Deng et al., 2025) | 0.36 | 20.46 | 0.04 |
| UDM-GRPO | **0.96** | **23.81** | 0.57 |

*Table 3.* Ablation study of different methods for integrating GRPO into our base model on GenEval, PickScore and OCR. The best and second-best results are indicated by **bold** and underlined, respectively.

| Model | Action | Trajectory | GenEval | PickScore | OCR |
|---|---|---|---|---|---|
| URSA | - | - | 0.69 | 21.79 | 0.08 |
| URSA | $x_1^t$ | backward | 0.84 | 21.99 | 0.23 |
| URSA | $\hat{x}_1$ | backward | 0.89 | 23.10 | 0.23 |
| URSA | $\hat{x}_1$ | forward | 0.94 | 23.51 | 0.34 |
| URSA (w/o CFG) | $\hat{x}_1$ | forward | **0.96** | **23.81** | **0.57** |

surpassing both prior RL methods and pre-trained baselines across a range of model variants and sizes, establishes a new state-of-the-art under both continuous and discrete settings.

Figure 5 presents qualitative comparisons of T2I generation among SD3.5, FLUX, URSA, and our method. After RL post-training, the URSA model exhibits substantial improvements in spatial arrangement, attribute binding, and object counting on the compositional prompts of GenEval. On the trival prompts of PickScore (Kirstain et al., 2023), UDM-GRPO produces images with finer details and fewer artifacts, preserving style consistency. These improvements highlight the effectiveness of our approach in enhancing

visual fidelity and text-image alignment for UDM models.

We further evaluate our approach on two downstream tasks: text rendering and human preference alignment. As shown in Table 2, UDM-GRPO achieves the state-of-the-art performance on PickScore. For text rendering, although the pre-trained model exhibits poor ocr performance, UDM-GRPO still achieves a substantial improvement.

### 5.3. Reduced-Step Accelerated Optimization

In this section, we investigate the strategies for few-step optimization to demonstrate the efficiency of our approach. We evaluate three timestep selection strategies: (1) **Early high-noise timesteps**: optimizing the three consecutive timesteps from the first half of the diffusion timestep; (2) **Random consecutive timesteps**: randomly sampling the three consecutive timesteps; and (3) **All timesteps**: optimizing over the entire diffusion timesteps.

We conduct head-to-head comparisons of the three strategies described above on GenEval, PickScore, and OCR tasks. As shown in subplots (d1, d2, d3) of Figure 6, optimizing the early high-noise timesteps yields a clear advantage on GenEval and PickScore, while the performance differences remain minor for OCR. These results demonstrate both the efficiency and effectiveness of few-step training approach.

### 5.4. Ablation Study

**Action Choice: Final Clean Sample $\hat{x}_1$ vs. Intermediate Predicted Sample $x_1^t$.** We study how the *action parameterization* affects backward optimization. We compare two choices: (i) using the final denoised output at $t = 1$ as the

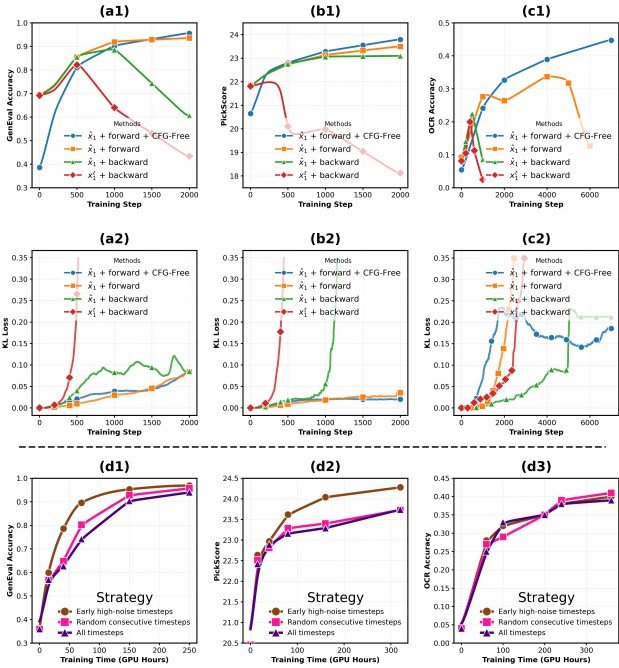

Figure 6. **Experimental results.** Performance metrics and KL loss on GenEval (a1, a2), PickScore (b1, b2), and OCR (c1, c2). The effects of different timestep optimization strategies across tasks are shown in (d1, d2, d3).

action, $a = \hat{x}_1$, and (ii) using the model's step-$t$ estimate of the final output as the action, $a = x_1^t$ for $t \in (0, 1)$. Starting from the original Flow-GRPO-based formulation, we replace the intermediate-prediction action (red) with the final-sample action (green) and perform head-to-head comparisons on GenEval, PickScore, and OCR in Figure 6 and Table 3. Using $a = \hat{x}_1$ consistently improves training performance on GenEval and PickScore, and leads to a lower KL divergence to the reference policy, suggesting more stable optimization. The gain is less pronounced on OCR, indicating that the underlying challenge for text rendering remains.

**Trajectory Choice: Forward vs. Backward.** We compare forward and backward optimization, which differ only in the construction of the state $x_t$. Specifically, forward optimization resamples $x_t$ from the forward diffusion process, whereas backward optimization uses intermediate denoising states from the reverse process. The action is defined as the clean sample in both settings. As shown in Figure 6 and Table 3, although both methods exhibit similar performance in the early training stage, backward optimization (green) later suffers from slow convergence and eventually collapses when the GenEval score reaches approximately 0.89. In contrast, forward optimization (orange) improves smoothly and stably, reaching a score of 0.95. Moreover, forward optimization consistently yields lower KL divergence, indicating superior training stability. Similar performance trends are also observed on the other two tasks.

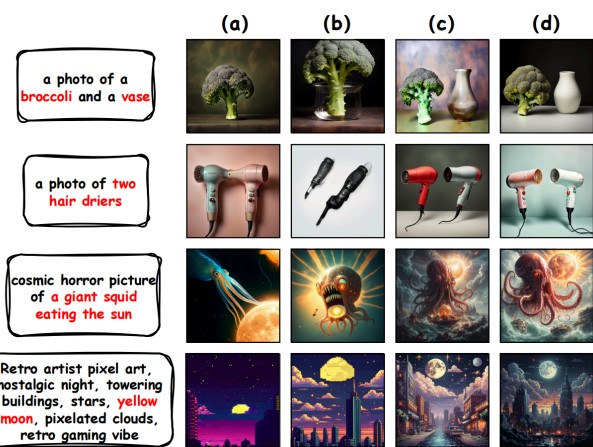

Figure 7. Qualitative Comparison. We compare different methods for integrating GRPO into our base model. From left to right, the results correspond to (a): backward + $x_1^t$, (b): backward + $\hat{x}_1$, (c): forward + $\hat{x}_1$, and (d): forward + $\hat{x}_1$ + CFG-free.

**CFG-free. vs. CFG.** We compare forward optimization with and without classifier-free guidance (CFG). As shown in Figure 6 and Table 3, although the CFG-Free (blue) setting performs poorly early in training due to lower sample quality, it surpasses the CFG-based (orange) training as optimization progresses. Overall, CFG-Free converges faster and achieves a lower KL divergence, indicating improved training efficiency and stability. Notably, on the OCR task, CFG-Free outperforms the CFG-based setting, suggesting that removing CFG broadens the policy distribution and enables more effective exploration during RL training.

**Qualitative Result.** We also provide visualizations corresponding to the ablation experiments described above, as shown in Figure 7. From left to right, the results correspond to: backward + $x_1^t$, backward + $x_1$, forward + $\hat{x}_1$, and forward + $\hat{x}_1$ + CFG-Free. It can be observed that our final method demonstrates clear advantages over the previous approaches.

## 6. Conclusion

In this paper, we propose **UDM-GRPO**, the first method that integrates the Uniform Discrete Diffusion Model with GRPO for text-to-image generation. By treating the final clean sample as the action and reconstructing the trajectory through the forward diffusion process, our method effectively addresses the instability caused by naive adaptation. Furthermore, we enhance training efficiency via Reduction-Step and CFG-free strategy. **UDM-GRPO** significantly improves the performance of the base model across multiple T2I tasks. In future work, we will investigate extending our framework to text-to-video generation and to more challenging multi-reward optimization task.

## Impact Statement

This paper presents work whose goal is to advance the field of Machine Learning. There are many potential societal consequences of our work, none which we feel must be specifically highlighted here.

## Acknowledgement

This work was supported by the Hainan Provincial Joint Project of Li'an International Education Innovation Pilot Zone (Grant No.624LALH008), BUPT Kunpeng&Ascend Center of Cultivation, NSFC (Grant No.61601042), and the Super Computing Platform of BUPT.

We would like to acknowledge Shilin Lu and Yuanzhi Zhu for the insightful discussions. We are grateful to Jiazhen Yan, Junwei Liu, Yuanyuan Li, Shaqi Luo, and Shuchen Weng for their significant support to this work. We also thank Yuanzhi Zhu, Zhipeng Chen, Jing Zuo, and Hongcan Xiao for their valuable feedback on the draft.

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

## Appendix

In this appendix, implementation details, experiments, and qualitative results are organized as follows:

- Training Details (A)
- Distribution Analysis (B)
- Extended Experimental Results (C)

## A. Training Details

### A.1. Pseudo Code for UDM-GRPO

We present the detailed pseudo code of the proposed UDM-GRPO in Algorithm 1.

---

**Algorithm 1** UDM-GRPO

---

1: **Input:** KL weight $\beta$, clip parameter $\epsilon$, reference policy $\pi_{\text{ref}}$, candidate timestep groups $\mathcal{T}_{\text{group}}$
2: **Initialize:** $\theta \leftarrow \theta_{\text{old}}$
3: **for** each iteration $n = 1, 2, \ldots$ **do**
4:     **for** each prompt $c \sim \mathcal{C}$ **do**
5:         Sample $G$ trajectories $\{\tau^i\}_{i=1}^{G} \sim \pi_{\theta_{\text{old}}}(\cdot \mid c)$                                     *// CFG-Free*
6:         Extract clean samples $\{\hat{x_1}^i\}_{i=1}^{G}$ and advantages $\{\hat{A}_i\}_{i=1}^{G}$
7:         Sample candidate timestep groups $\{(t_i^1, t_i^2, t_i^3)\}_{i=1}^{G} \sim \mathcal{T}_{\text{group}}$               *// Reduction-Step*
8:     **end for**
9:     Initialize total loss $\mathcal{L} \leftarrow 0$
10:     **for** $i = 1, \ldots, G$ **do**
11:         **for** $j = 1, 2, 3$ **do**
12:             Sample noisy state $\hat{x}_{t_i^j}^i \sim p_{t_i^j}(x \mid \hat{x}_1^i)$                                   *// Forward-Process*
13:             $r_{t_i^j}^i(\theta) \leftarrow \dfrac{p_\theta(\hat{x}_1^i \mid \hat{x}_{t_i^j}^i, c)}{p_{\theta_{\text{old}}}(\hat{x}_1^i \mid \hat{x}_{t_i^j}^i, c)}$                  *// Accurate Action Strategy*
14:             $\mathcal{J}_{\text{policy}}^{(t_i^j, i)} \leftarrow \min\left(r_{t_i^j}^i(\theta)\hat{A}_i, \text{clip}(r_{t_i^j}^i(\theta), 1 - \epsilon, 1 + \epsilon)\hat{A}_i\right)$
15:             $\mathcal{L} \leftarrow \mathcal{L} - \mathcal{J}_{\text{policy}}^{(t_i^j, i)} + \beta\, D_{\text{KL}}\left(p_\theta(\cdot \mid \hat{x}_{t_i^j}^i, c)\,\|\,p_{\text{ref}}(\cdot \mid \hat{x}_{t_i^j}^i, c)\right)$
16:         **end for**
17:     **end for**
18:     $\theta \leftarrow \theta - \lambda \nabla_\theta \mathcal{L}$                                                          *// Policy Optimization*
19:     $\theta_{\text{old}} \leftarrow \theta$
20: **end for**

---

## B. Distribution Analysis

In Section 4.2, we validate that $\mathcal{X}_{\text{forward}}$ is closer to the forward trajectory than $\mathcal{X}_{\text{backward}}$ by comparing the FID between each trajectory and $\mathcal{X}_{\text{pretrain}}$, as well as through visual comparisons. In this section, we provide a detailed description of the experimental setup and computation procedure.

Specifically, we first sample 2,048 pairs of captions and corresponding images $(c, x_1)$ from the URSA pretraining dataset. Following the trajectory definitions in Section 3.1, we generate three types of trajectories for each pair: the forward trajectory $\mathcal{X}_{\text{forward}}$, the backward trajectory $\mathcal{X}_{\text{backward}}$, and the pretraining trajectory $\mathcal{X}_{\text{pretrain}}$. For each trajectory, we sample $x_1^t$ from the conditional distribution $p_\theta(x_1^t \mid x_t)$ given $x_t$. The resulting index predictions are subsequently converted to image space using the model's standard decoding procedure. To quantitatively measure how closely each trajectory aligns with the pretraining trajectory $\mathcal{X}_{\text{pretrain}}$, we compute the Fréchet Inception Distance (FID) between the distributions of predicted images at the same timestep. Specifically, for a given timestep $t$, we treat the set of predicted images from $\mathcal{X}_{\text{forward}}$ and $\mathcal{X}_{\text{backward}}$ as two empirical distributions and compute their FID with respect to $\mathcal{X}_{\text{pretrain}}$. All FID computations are performed using the official PyTorch implementation, ensuring consistency with standard evaluation practices.

# C. Extended Experimental Results

In this section, we conduct additional experiments to systematically demonstrate the effectiveness of our method from multiple perspectives.

## C.1. Generalized Validation

To further validate the generality of our method, we conduct additional experiments on FUDOKI(Wang et al., 2025a), a UDM-based multimodal large language model that unifies visual understanding and image generation. We adopt the same experimental setup as used for URSA and conduct training across three benchmarks—GenEval, PickScore, and OCR—using 15 inference steps for group sampling and 32 inference steps for evaluation. As shown in the Table 4, UDM-GRPO consistently and significantly improves the performance of the original model. These findings demonstrate that our method is not only effective for standalone T2I models, but also generalizes well to MLLMs, further confirming its broad applicability.

*Table 4.* Performance of FUDOKI with UDM-GRPO on GenEval, PickScore and OCR tasks.

| Model | #Params | PickScore | OCR | GE(Overall) | GE(Single Obj.) | GE(Two Obj.) | GE(Counting ) | GE(Colors) | GE(Position) | GE(Attr. Binding) |
|---|---|---|---|---|---|---|---|---|---|---|
| FUDOKI | 1.5B | 21.32 | 0.04 | 0.76 | 0.96 | 0.86 | 0.51 | 0.90 | 0.67 | 0.64 |
| FUDOKI (w/ UDM-GRPO) | 1.5B | **23.40** | **0.26** | **0.86** | **0.99** | **0.90** | **0.89** | **0.99** | **0.87** | **0.72** |

## C.2. Different Model Performance Comparison

We provide additional visualizations comparing the baseline, our method, SD3.5-L, and Flux.1 Dev to further highlight the superior performance of our approach as shown in Figure 8. In particular, our model demonstrates enhanced generative capacity, including the ability to accurately model complex scenes with a larger number of objects and to produce outputs that better align with human perceptual quality metrics. These results further illustrate the potential of UDM-GRPO to augment the capabilities of the base model.

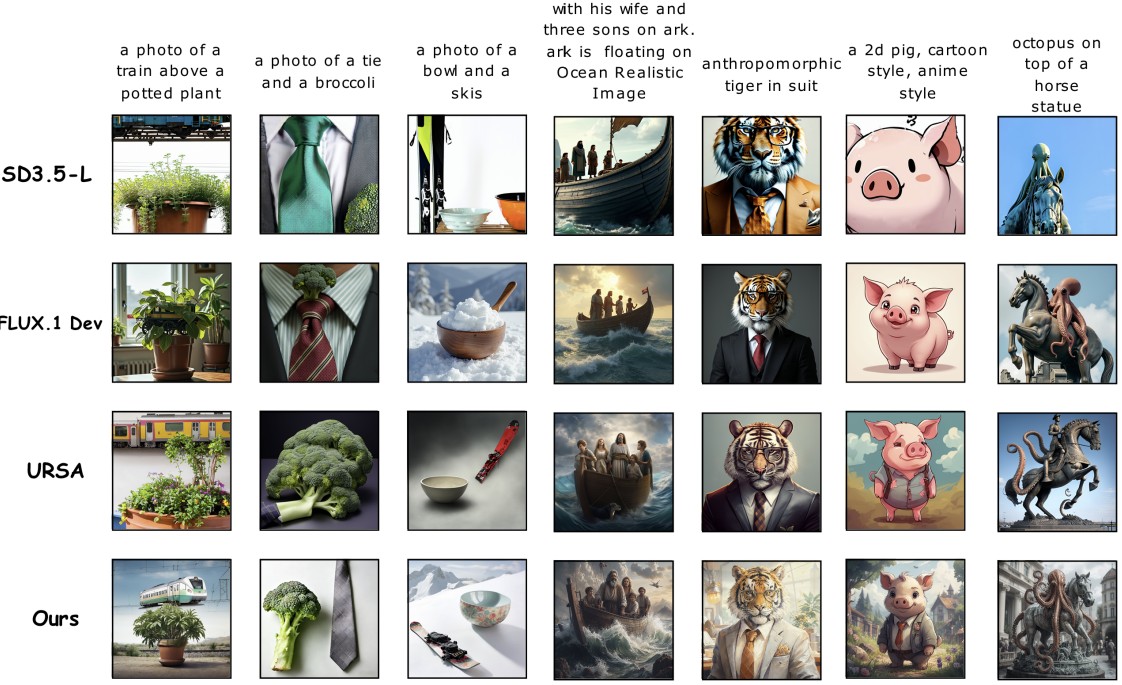

*Figure 8.* Qualitative Comparison. The prompts are taken from GenEval, PickScore respectively, where we compare the SD3.5-L and Flux.1 Dev with our model.

## C.3. Different Method Qualitative Comparision

In this section, we present the results of models trained with different methods, as discussed in Section 5, thereby enabling a comprehensive comparison of their performance. As shown in Figures 9, the methods from left to right correspond to backward + $x_1^t$, backward + $x_1$, forward + $\hat{x}_1$, and forward + $\hat{x}_1$ + CFG-free. We observe that the initial integration with GRPO already improves the model's capabilities, while the addition of Accurate Action and the forward strategy further improves performance. Moreover, the results indicate that CFG does not compromise generative quality, demonstrating the model's strong generative capacity.

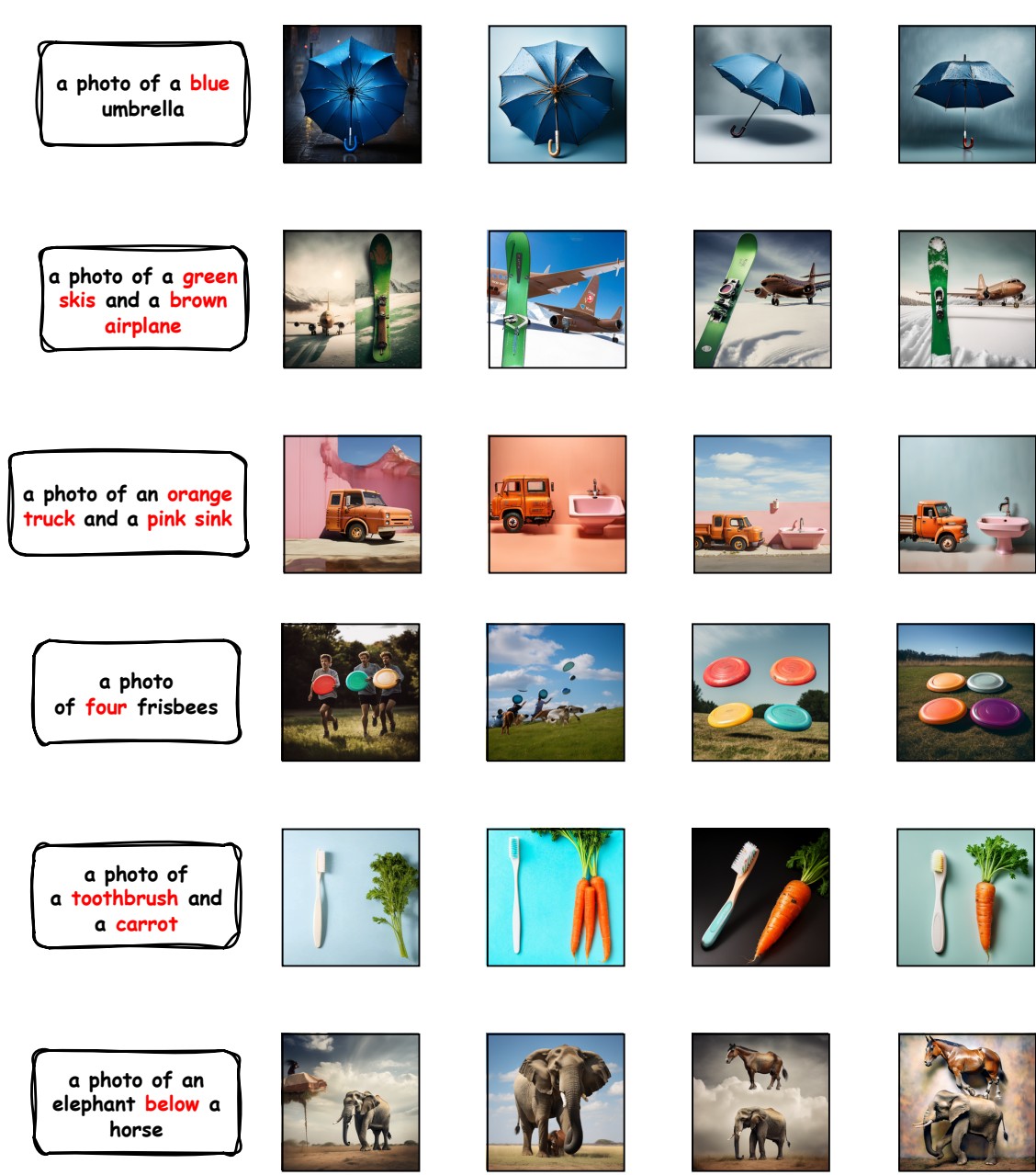

*Figure 9.* Visualization for different method.

### C.4. Training Process

To better understand the training dynamics of our UDM-GRPO framework, we visualize the evolution of generated samples corresponding to fixed evaluation prompts at regular intervals during training. As shown in Figure 10, it is evident that the quality of generated samples improves progressively over time, with their accuracy steadily increasing.

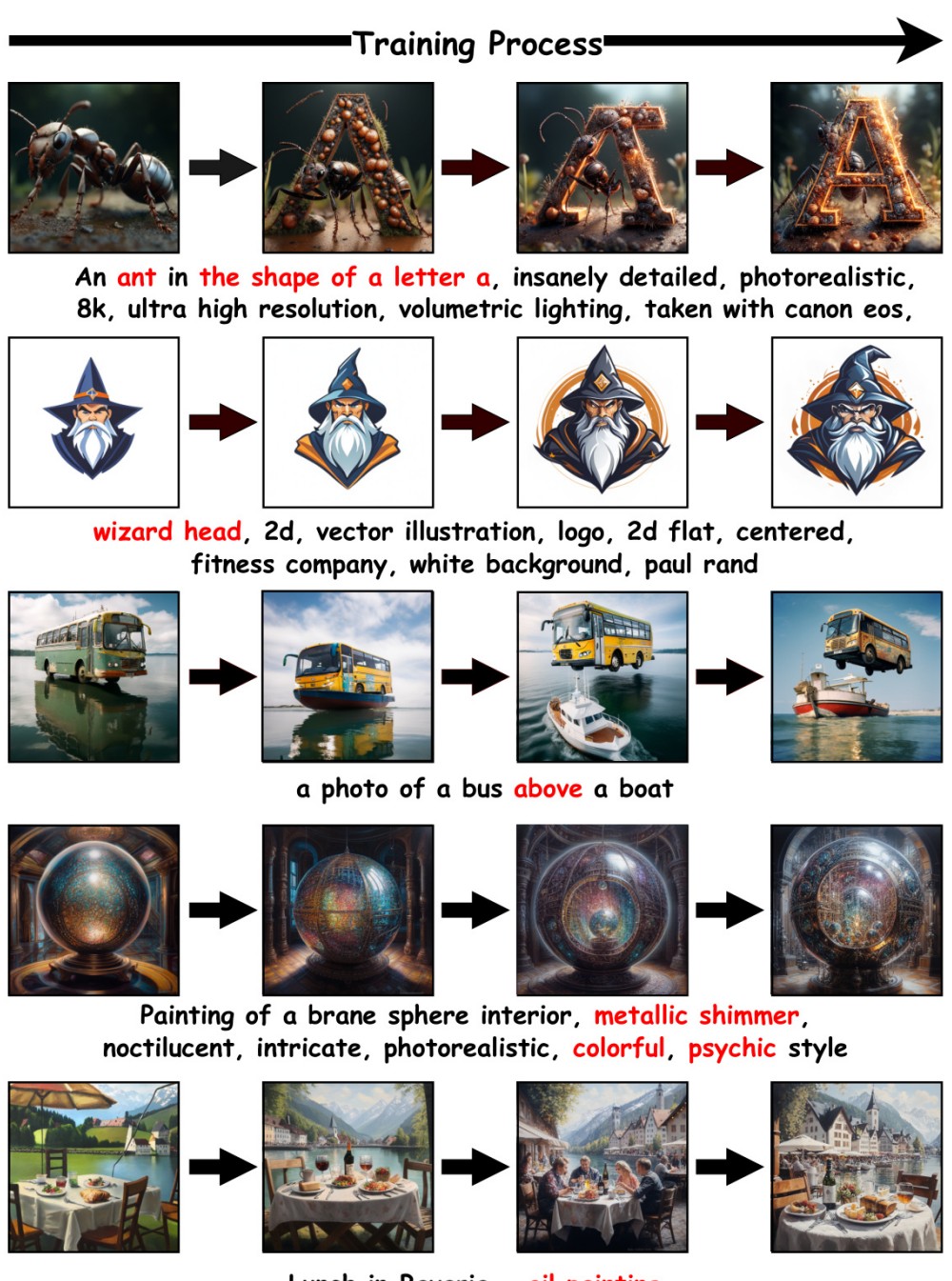

*Figure 10.* We visualize the generated samples across successive training iterations during the optimization.

