# OpenReview forum: "UDM-GRPO: Stable and Efficient Group Relative Policy Optimization for Uniform Discrete Diffusion Models"
_ICML.cc/2026/Conference — ICML 2026 spotlight_

### Official Review · Reviewer_J39C · 2026-02-26

**Soundness:** 3
**Presentation:** 2
**Significance:** 2
**Originality:** 3
**Overall Recommendation:** 4
**Confidence:** 3

**Summary:**

The authors find limitations in integrating Group Relative Policy Optimization (GRPO) with Uniform Discrete Diffusion Models (UDM) for text-to-image generation. The authors investigate a significant issue rooted in two problems: inaccurate intermediate actions (high-entropy, erroneous early-step predictions are treated as optimization signals, leading the model to reinforce misleading information) and biased backward trajectory distribution (reverse-process sampling states create a distribution mismatch with pretraining’s forward noising process, resulting in out-of-distribution training). This paper seeks to investigate a central concept of aligning RL optimization with UDM’s pretraining objectives and processes, proposing UDM-GRPO with two core strategies (redefining policy actions as the final clean sample, reconstructing training trajectories via the forward diffusion process) and two efficiency-boosting training tactics (Reduction-Step optimization concentrating on high-noise timesteps, CFG-Free training decoupling conditional and unconditional forward passes). The experiments show the effectiveness of the method.

**Compliance With Llm Reviewing Policy:**

Affirmed.

**Final Justification:**

Thanks for the authors' response. It has addressed my concerns, and I will maintain my score.

**Key Questions For Authors:**

1. Can the authors provide rigorous theoretical justifications (e.g., mathematical proofs, formal analysis of optimization stability/convergence, or theoretical bounds on performance gains) to validate the inherent superiority of the two core insights (defining the final clean sample as the policy action and reconstructing training trajectories via the forward diffusion process) for UDM-GRPO, beyond the empirical experimental evidence presented in the paper?
2. In Table 1, the authors fail to conduct a controlled comparison that keeps the base UDM model unchanged and contrasts it with other GRPO-based methods. This lack of a homogeneous baseline comparison makes it difficult to directly quantify the unique performance gains of UDM-GRPO over existing GRPO variants for discrete generative models.

**Limitations:**

yes

**Strengths And Weaknesses:**

Strengths:
1. The paper features a clear and logical structure, with sufficient motivation for addressing the critical gap of RL integration with UDM and rigorous analysis of the root causes of training instability in naive GRPO adaptation.
2. Intuitive visualizations (e.g., reward-KL training curves) effectively support the research motivation, method design, and experimental conclusions, making the key findings easy to comprehend.

Weaknesses:
1. The study lacks the reporting of fundamental and widely-recognized generative model evaluation metrics (e.g., FID, CLIP-Score, IS, SSIM), which are essential for comprehensively assessing the quality, diversity, and text-image alignment of generated samples.
2. No efficiency comparisons with baselines are provided, including runtime, GPU memory and parameter count.
3. The paper contains minor typographical errors in key technical terms and writing (e.g., the misspelling "optimizaiton" for "optimization" in the contributions section; misspelled "Visualazition" in appendix figure captions).

---

> ### Author Rebuttal · Authors · 2026-03-31
>
> # **Q1: The study lacks more generative model evaluation metrics.**
> Thank you for the insightful suggestion. We further evaluate models trained on three tasks across three dimensions: **(1) image quality: Aesthetic Score[1] ; (2) preference score: ImageReward[2]; and (3) text–image alignment: CLIP-Score**. All evaluation prompts are sourced from the DrawBench[3].
> As shown in Table 1, these results demonstrate that **UDM-GRPO improves both task performance and the overall quality of generated images**.
>
> **Table1** Evaluation Results.
> | Model | GenEval | OCR | PickScore | Aesthetic | ImageReward | CLIP-Score |
> |:-----|:------:|:---:|:---------:|:--------:|:-----------:|:----------:|
> | URSA | 0.69 | 0.08 | 21.79 | 5.36 | 0.67 | 26.9 |
> | UDM-GRPO | 0.96 | - | - | 5.43 | 0.83 | **27.24** |
> | UDM-GRPO | - | 0.57 | - | 5.35 | 0.68 | 27.21 |
> | UDM-GRPO | - | - | 23.81 | **6.39** | **1.18** | 27.11 |
>
> # **Q2: No efficiency comparisons with baseline.**
> We thank the reviewer for pointing out this important aspect. Results for the GenEval task are presented in Table 2. Both methods fine-tune URSA-1.7B and have the same parameter count, but the baseline stores all sampled results, **resulting in higher GPU memory usage**. UDM-GRPO reconstructs trajectories via $x_t$ resampling, with **a modest increase in per-epoch runtime**. Nevertheless, UDM-GRPO demonstrates clear improvements in both training stability and overall efficiency (Figure 6 in paper).
>
> **Table2**  Efficiency Comparisons.
> | Model | Latency(epoch/s) | Memory(GB) | Params(B) | GenEval|
> |:-----|:------:|:---:|:---------:| :---------:|
> | baseline | 15.84 | 37.8G | 1.7B | 0.84|
> | UDM-GRPO | 16.15 | 37.1G | 1.7B| 0.96|
> # **Q3: The paper contains minor typographical errors in key technical terms and writing.**
> We thank the reviewer for pointing out the mistakes. We have made modifications in the manuscript.
>
> # **Q4: Can the authors provide rigorous theoretical justifications?**
> We thank the reviewer for this constructive comment. We provide the theoretical justification below :
>
> **Proposition I.**   Using $x_1$ as the action reduces gradient variance compared to using $x_1^t$.
>
> **Proof.**
> The policy gradient for two method can be expressed as:
>
> $\nabla_\theta J_{\text{baseline}} \approx \mathbb{E}_{t, x_t} \Big[ \nabla\_\theta \log p\_\theta(x^t_1 \mid x_t) \\hat{A} \Big] $
>
> $\nabla_\theta J_{\text{UDM-GRPO}} \approx \mathbb{E}_{t, x_t} \Big[ \nabla\_\theta \log p\_\theta(x_1 \mid x_t) \hat{A} \Big] $
>
> where $x^t_1 = x_1 + \epsilon_\theta(x_t, t)$ and $\epsilon_\theta$ denotes the prediction error.
>
> In the baseline, $x^t_1$ is used as the action: $\nabla\_\theta \log p_\theta(x^t_1 \mid x_t)
> = \nabla_\theta \log p_\theta(x_1 + \epsilon_\theta \mid x_t)$.
>
> Since $\epsilon\_\theta$ has non-zero variance, the gradient variance is amplified:
>
> $\mathrm{Var}\big[\nabla\_\theta \log p_\theta(x^t_1 \mid x_t)\big] = \mathrm{Var}\big[\nabla_\theta \log p_\theta(x_1 \mid x_t) + \text{noise term}\big] > \mathrm{Var}\big[\nabla_\theta \log p_\theta(x_1 \mid x_t)\big]$.
>
> **Proposition II.**  Resampling $x_t \sim q(x_t \mid x_1)$ stabilizes the GRPO ratio.
>
> **Proof.**
> Sampling from an imperfect policy produces $x_t^{gen}$ that drifts away from the pretrain data manifold $\mathcal{M}$, i.e., $x_t^{gen} \notin \mathrm{Support}(\mathcal{M})$, which leads to $p_\theta(x_1 \mid x_t^{gen}) \to 0$,
> thereby causing the importance ratio to explode.
>
> By instead resampling $x_t \sim q(x_t \mid x_1)$, we enforce $x_t \in \mathrm{Support}(\mathcal{M}),$ ensuring that $p_\theta(x_1 \mid x_t)$ remains non-vanishing. This keeps likelihood values well-behaved and prevents instability caused by distribution mismatch.
>
> # **Q5: Fail to conduct a controlled comparison in Table1.**
> We appreciate the reviewer’s suggestion.  As shown in Table3, we observe that the baseline achieves notable improvements only on the Single Object and Position. In contrast, correcting the action to $x_1$ leads to **further gains on Counting**. Moreover, updating the trajectory to the forward process **improves performance on Color and Attribute Binding, resulting in substantial improvements across all metrics**.
>
> **Table3** GenEval Results.
> | **Model** | **Overall** | **Single** | **Two** | **Count** | **Color** | **Position** | **Attr. Binding** |
> | :--- | :---: | :---: | :---: | :---: | :---: | :---: | :---: |
> | URSA | 0.69 | 0.99 | 0.91 | 0.60 | 0.87 | 0.28 | 0.49 |
> | $x^t_1$ + backward(baseline)| 0.84 | 0.98 | 0.94 | 0.79 | 0.82 | 0.85 | 0.64 |
> | $x_1$ + backward| 0.89 | 1.00 | 0.99 | 0.90 | 0.87 | 0.95 | 0.65 |
> | UDM-GRPO| **0.96** | **1.00** | **1.00** | **0.97** | **0.97** | **0.98** | **0.82** |
>
> [1] Chrisoph Schuhmann. Laion aesthetics, Aug 2022.
>
> [2] Xu J, Liu X, Wu Y, et al. Imagereward: Learning and evaluating human preferences for text-to-image generation. NeurIPS 2023.
>
> [3] Saharia C, Chan W,  et al. Photorealistic text-to-image diffusion models with deep language understanding. NeurIPS 2022.

---

> > ### Author Rebuttal · Reviewer_J39C · 2026-04-01
> >
> > Thanks for the authors' response. It has addressed my concerns, and I will maintain my score.

---

> > > ### Author Response · Authors · 2026-04-05
> > >
> > > Dear reviewer **J39C**,
> > >
> > > Thank you for the time you've invested in reviewing our work. Your encouragement is greatly appreciated!
> > >
> > > Best regards,
> > >
> > > Authors

---

### Official Review · Reviewer_D233 · 2026-03-05

**Soundness:** 1
**Presentation:** 3
**Significance:** 2
**Originality:** 2
**Overall Recommendation:** 4
**Confidence:** 3

**Summary:**

This paper adapts GRPO to finetuning uniform discrete models for text-to-image translation. After identifying instability issues in naive implementation, the paper proposes a modified MDP formulation, termed UDM-GRPO, which shows improved stability and performance.

**Compliance With Llm Reviewing Policy:**

Affirmed.

**Final Justification:**

The authors have addressed my concerns, and I have raised my score.

**Key Questions For Authors:**

Please see the Weakness section.

**Limitations:**

The authors are suggested to provide potential negative societal impact in the field of text-to-image generation.

**Strengths And Weaknesses:**

Strengths:

1. The paper provides a novel way to integrate uniform discrete model with GRPO

2. The paper presents some good experimental results and performs sufficient ablation studies

Weaknesses:

1. Restricted to particular application: The methodology used in this paper seems quite ad hoc for text-to-image. There is no indication for wider applicability.

2. Motivation for flow model unclear: Rather than uniform diffusion models, the paper is using the flow model with uniform source distribution instead. This choice is very weird. The flow model is more commonly used when the source distribution is arbitrary, in which case one learns the CTMC rate. If we restrict to uniform source, it is more natural to use the uniform diffusion with known forward CTMC rate and with a different loss function.

3. Atypical sampling process: The paper assumes a two-step sampling scheme where $\hat x_1$ is first sampled (which somehow contributes to the instability). This approach, however, is not common in either the discrete flow or uniform diffusion models. Instead, one uses Bayesian rule and sums over all possible $\hat x_1^i$ to get the probability to sample for $x_{t+\Delta t}^i$ (e.g., see (Campbell et al., 2024)).

4. Unclear numerical advantage: The results in Table 1 does not really show the benefit of using discrete models rather than continuous flow, not to mention that additional care for stabilization is needed in this case.

5. Missing comparison: There lacks any comparison with Flow-GRPO in Table 2.

---

> ### Author Rebuttal · Authors · 2026-03-31
>
> # **Q1: Restricted to particular application.**
>
> Thank you for the feedback. We choose text-to-image (T2I) generation as our experimental task, as it is a representative problem in visual generation and the focus of prior RL-based generative studies [6, 7]. It should be noted that our core contribution addresses inaccurate actions and trajectory issues when applying RL to UDM.
>
>
> As shown in Table 1, to support generality, we further **validate UDM-GRPO on FUDOKI-1.5B**, a UDM-based multimodal large language model (MLLM) that unifies visual understanding and image generation, **demonstrating its effectiveness across different model architectures**.
>
> **Table1**. UDM-GRPO Performance on FUDOKI.
> | **Model** | **PickScore** | **OCR** | **GE(Overall)** | **GE (Single)** | **GE (Two)** | **GE (Count)** | **GE (Color)** | **GE (Pos)** | **GE (Attr)** |
> | :--- | :---: | :---: | :---: | :---: | :---: | :---: | :---: | :---: | :---: |
> | FUDOKI | 21.32 | 0.04 | 0.76 | 0.96 | 0.86 | 0.51 | 0.90 | 0.67 | 0.64 |
> | **FUDOKI (w/ UDM-GRPO)** | **23.40** | **0.26** | **0.86** | **0.99** | **0.90** | **0.89** | **0.99** | **0.87** | **0.72** |
>
>
>
>
> # **Q2: Motivation for flow model unclear.**
>
> We adopt Flow Matching, which has become **a dominant paradigm in image generation and is widely recognized for its advantages in generation quality, sampling speed, and stability [1, 2, 3, 4, 5, 6].** In fact, **Diffusion and Flow Matching are increasingly viewed as a unified generative paradigm**, with largely shared downstream methodologies. **As noted in [2], Flow Matching can be regarded as a velocity-field extension of the Diffusion framework. Furthermore, [3, 4] explicitly demonstrate that their methods belongs to the discrete diffusion paradigm**. This is why, under a uniform initial distribution, such models are referred to as Uniform Discrete Diffusion Models (UDM).
>
>
> # **Q3: Atypical sampling process.**
>
> Thank you for the comment. We would like to clarify that the two-stage Euler Solver is not atypical — it is the standard inference procedure introduced and thoroughly validated in [5], which is the foundational work for UDM.
>
> The reviewer suggests marginalizing over all $x_1$ to compute distribution of $x_{t +\Delta t}$. **While theoretically sound, computing it scales as $|\mathcal{T}|^2$ per token, making it computationally infeasible for our setting with a vocabulary size of $|\mathcal{T}| = 131{,}072$ and sequence length $D = 1024$.** The two-stage Euler solver is specifically designed for this large-vocabulary regime, enabling efficient sampling (14.13s per image), whereas the marginal approach would be intractable.
>
>
> **Table2**. URSA Runtime and GPU Peak.
> | Method | Denoise Step | Time(s) | GPU Peak(G) |
> | :--- | :---: | :---: | :---: |
> | Euler Solver | 25 | 14.13 | 6.87 |
>
>
> # **Q4: Unclear numerical advantage & Q5: Missing comparison with Flow-GRPO.**
>
> We thank the reviewer for the two questions. **We would like to clarify that our goal is not to compare discrete and continuous models, but to address a key open problem: stabilizing RL for UDM.** To our knowledge, UDM-GRPO is the first method to achieve stable RL training in this setting.
>
> We aim to highlight that **UDM-GRPO achieves clear advantages over Flow-GRPO in both performance and convergence efficiency**. As shown in Table 3, **UDM-GRPO not only outperforms Flow-GRPO but also requires significantly fewer GPU Hours**. For instance, UDM-GRPO achieves **GenEval 0.95 with a 10.2× speed-up**, and **PickScore 23.10 with an 18.2× speed-up**.  Furthermore, it reaches a **GenEval score of 0.96 and a PickScore of 23.81**, establishing **state-of-the-art** performance.
>
>
> **Table3**. Performance Comparison. Speed-up is Relative to Flow-GRPO Baseline.
> | Method | GenEval  | GenEval GPU Hours | PickScore | PickScore GPU Hours |
> | :--- | :---: | :---: | :---: |:---: |
> | SD3.5-M | 0.71 | - | 21.72 | - |
> | SD3.5-M (w/Flow-GRPO) | 0.95 | 2223.4 (1×) |23.10 | 950.6 (1×) |
> | URSA | 0.69 | - | 21.79 | -|
> | URSA(w/UDM-GRPO) | 0.95 | 217.6 (10.2×)| 23.10 | 52.1(18.2×) |
> | URSA(w/UDM-GRPO) | **0.96** | 256.2 | **23.81** | 210.3 |
>
> [1] Wang J, Lai Y, Li A, et al. Fudoki: Discrete flow-based unified understanding and generation via kinetic-optimal velocities. NeurIPS 2025.
>
> [2] Zhengyang Geng, Kaiming He, et al. Mean flows for one-step generative modeling. NeurIPS 2025.
>
> [3] Itai Gat, Tal Remez, Neta Shaul, et al. Discrete flow matching. NeurIPS 2024.
>
> [4] Deng H, Pan T, Zhang F, et al. Uniform discrete diffusion with metric path for video generation. ICLR 2026.
>
> [5] Shaul N, Gat I, Havasi M, et al. Flow matching with general discrete paths: A kinetic-optimal perspective. ICLR 2025.
>
> [6] Liu J, Liu G, Liang J, et al. Flow-grpo: Training flow matching models via online rl. NeurIPS 2025.
>
> [7] Luo Y, Hu X, Fan K, et al. Reinforcement learning meets masked generative models: Mask-grpo for text-to-image generation. NeurIPS 2025.

---

> > ### Author Rebuttal · Reviewer_D233 · 2026-04-01
> >
> > The authors have fully addressed my concerns. I have raised my score.

---

> > > ### Author Response · Authors · 2026-04-05
> > >
> > > Dear reviewer **D233**,
> > >
> > > It is encouraging to see that the issues you raised have now been fully addressed! Thank you for your time, effort, and valuable feedback.
> > >
> > > Best regards,
> > >
> > > Authors

---

### Official Review · Reviewer_MAqv · 2026-03-15

**Soundness:** 3
**Presentation:** 3
**Significance:** 2
**Originality:** 2
**Overall Recommendation:** 5
**Confidence:** 2

**Summary:**

This paper proposes UDM-GRPO, a reinforcement learning post-training method for UDMs that introduces final-image actions and forward trajectory reconstruction to make GRPO more stable in the UDM setting. Experiments show that, compared with more naive backward-based designs, the proposed method improves both training stability and generation quality on benchmarks such as GenEval, PickScore, and OCR.

**Compliance With Llm Reviewing Policy:**

Affirmed.

**Final Justification:**

The rebuttal provided additional data on the FUDOKI-1.5B model and further ablations, which resolved the concerns regarding generalizability and the specific role of the CFG-Free strategy. As the proposed UDM-GRPO framework is technically sound and the claims are adequately supported, I recommend accepting this paper.

**Key Questions For Authors:**

The authors should clarify and respond to the points raised in the Weaknesses section.

**Limitations:**

yes

**Strengths And Weaknesses:**

Strengths:
- The paper is well-organized and easy to read.
- The paper makes a convincing case for why a straightforward application of a standard diffusion-RL method such as Flow-GRPO does not work well in the UDM setting, and clearly motivates the proposed reformulation from that failure mode. In particular, the idea of treating the final image as the action is well grounded in the generation mechanism of UDM, so this part of the method feels especially intuitive. In addition, the ablation study helps clarify how much each design choice contributes, which further strengthens the overall credibility of the approach.
- The paper also provides solid empirical evidence, showing improved training stability and stronger performance on GenEval, PickScore, and OCR, which supports the effectiveness of the proposed design choices.

Weakness:
- The method is validated only on a single UDM backbone, URSA-1.7B. While the empirical gains on this model are promising, it remains unclear whether the proposed method generalizes to other UDM backbones, model scales, or tokenization/model design choices. As a result, the paper does not yet establish whether the observed improvements are specific to URSA-1.7B or broadly applicable to discrete diffusion models.
- The paper mainly attributes its success to the final-image action formulation and forward trajectory reconstruction. However, Table 3 suggests that CFG-Free may account for a substantial portion of the overall improvement, especially on OCR, where the gain from adding CFG-Free is particularly large. This raises some ambiguity about which component is actually driving the final performance. Moreover, the reason why CFG-Free helps so much, especially for OCR, is not sufficiently analyzed beyond a tentative exploration-based explanation. It is also unclear whether this effect is specific to UDM/UDM-GRPO or would transfer more broadly to other diffusion-RL settings.

---

> ### Author Rebuttal · Authors · 2026-03-31
>
> # **Q1: Is UDM-GRPO exclusively designed for URSA, or can it be applied more generally?**
> Thank you for the insightful question. To further validate the generality of our method, we **conduct additional experiments on FUDOKI-1.5B[1], a UDM-based multimodal large language model (MLLM) that unifies visual understanding and image generation.**
>
> We adopt the same experimental setup as used for URSA and perform training across three benchmarks: GenEval, PickScore, and OCR. As shown in the results, **UDM-GRPO consistently and significantly improves the performance of the original model**. These findings demonstrate that our method is **not only effective for standalone T2I models, but also generalizes well to MLLMs, further confirming its broad applicability**.
>
> We sincerely appreciate your valuable feedback and will incorporate this result into the revised version of the paper.
>
> **Table1**. Performance of FUDOKI-1.5B with UDM-GRPO on GenEval, PickScore and OCR Tasks.
> | **Model** | **PickScore** | **OCR** | **GE(Overall)** | **GE (Single)** | **GE (Two)** | **GE (Count)** | **GE (Color)** | **GE (Pos)** | **GE (Attr)** |
> | :--- | :---: | :---: | :---: | :---: | :---: | :---: | :---: | :---: | :---: |
> | FUDOKI | 21.32 | 0.04 | 0.76 | 0.96 | 0.86 | 0.51 | 0.90 | 0.67 | 0.64 |
> | **FUDOKI (w/ UDM-GRPO)** | **23.40** | **0.26** | **0.86** | **0.99** | **0.90** | **0.89** | **0.99** | **0.87** | **0.72** |
>
> # Q2: Which component is primarily responsible for the final performance improvements: the two modification strategies or CFG-Free?
> Thanks for your comment.  We would like to clarify that **redefining the action as the final image and reconstructing the trajectory using the forward process** constitute the core contributions of our work, which are the **decisive factors for stabilizing training and achieving substantial performance gains**. In contrast, **adopting the CFG-Free strategy is primarily motivated by improving training efficiency**.
>
> As shown in Table2, the baseline (w/$x_1^t$, backward) shows an improvement in GenEval (0.69 → 0.84), but the reward gradually decreases afterwards, eventually approaching 0. However, we observe that: (i) replacing $x_1^t$ with $x_1$ already brings consistent improvements, and  (ii) replacing backward trajectories with forward further improves both stability and performance.  Notably, without these two components, training remains unstable, and CFG-Free alone cannot resolve this issue (Table2). **This indicates that the primary gains stem from addressing the two fundamental challenges in diffusion RL—action–reward misalignment and state distribution shift.** In Table3 (in the paper) for GenEval and PickScore tasks, URSA(w/ $x_1$, forward) and URSA (w/ $x_1$, forward, CFG-Free) are evaluated at the same training steps to compare efficiency, not their final converged performance.  Indeed, as shown in Table2, with sufficient training steps, using the standard CFG strategy can achieve performance comparable to CFG-Free.
>
> We acknowledge that the CFG-Free setting yields a more pronounced performance improvement on the OCR task. However, this effect arises primarily from the **limited capability of the underlying base model**. Specifically, the base model URSA achieves performance comparable to SD3.5 on GenEval and PickScore, but exhibits significantly lower initial performance on **OCR (URSA: 0.08 vs. SD3.5-M: 0.59)**. This indicates that the OCR task operates under a poor initialization scenario, where effective exploration is particularly critical for RL performance. In this context, CFG-Free introduces higher stochasticity and fewer constraints on the generation trajectory, thereby expanding the policy’s exploration space and enabling the model to escape suboptimal initial solutions and achieve better outcomes. However, for the GenEval and PickScore tasks, the CFG-Free strategy only improves convergence speed, without increasing the model’s ultimate performance ceiling. **Although this stems from the model’s performance limitations, it nonetheless provides a valuable insight: expanding the exploration space can be beneficial for reinforcement learning.**
>
>
> **Table2**:  Evaluation Results on GenEval.
> | Model | Action | Trajectory | Training Step | GenEval |
> |:-----|:------:|:---:|:------:|:---:|
> | URSA | - | - | - | 0.69 |
> | URSA | $x^t_1$ | backward | 500 | 0.84 |
> | URSA | $x^t_1$ | backward | 2000 | 0.42 |
> | URSA(w/ CFG-Free) | $x^t_1$ | backward | 500 | 0.62 |
> | URSA | $x_1$ | backward | 1000 | 0.89 |
> | URSA | $x_1$ | forward | 2000 | 0.94 |
> | URSA | $x_1$ | forward |   2500    | **0.96**|
> | URSA(w/ CFG-Free) | $x_1$ | forward | 2000| **0.96** |
>
> [1] Wang J, Lai Y, Li A, et al. Fudoki: Discrete flow-based unified understanding and generation via kinetic-optimal velocities. NeurIPS 2025.

---

> > ### Author Rebuttal · Reviewer_MAqv · 2026-04-04
> >
> > The additional experiments using the FUDOKI model and the detailed ablation data have completely resolved my concerns regarding the method's generalizability and the contribution of CFG-Free. I appreciate the convincing rebuttal, and I will raise my score.

---

> > > ### Author Response · Authors · 2026-04-05
> > >
> > > Dear reviewer **MAqv**,
> > >
> > > We are very glad that your concerns have been addressed! Thank you for taking the time to review our rebuttal and for the encouraging recommendation.
> > >
> > > Best regards,
> > >
> > > Authors

---

### Decision · Program_Chairs · 2026-04-30

**Decision:**

Accept (spotlight)

**Comment:**

This paper applies GRPO to uniform diffusion models with strong empirical results. At this time, many papers tend to focus either on pertaining or distillation but not reward based post-training. Therefore, this paper is very valuable for the ICML community. The paper would have gone beyond if it would have also considered text generation, another upcoming use case of uniform diffusion models.